# Exploring barriers to care home research recruitment during the COVID-19 pandemic: The influence of social media recruitment posts and public sentiment

Mariyana Schoultz[1]*, Claire Mcgrogan[1], Clare Carolan[2‡], Leah Macaden[3‡], Michelle Beattie[2‡]

1 School of Health and Life Sciences, Northumbria University, Newcastle Upon Tyne, United Kingdom,
2 Centre for Rural Health Sciences, University of the Highlands and Islands, Inverness, United Kingdom,
3 Nursing Studies, School of Health in Social Sciences, The University of Edinburgh, Edinburgh, United Kingdom

☯ These authors contributed equally to this work.
‡ These authors also contributed equally to this work
* mariyana.schoultz@northumbria.ac.uk

**Data Availability Statement:** Data are available from OSF at https://osf.io/f7sae/.

## Abstract

### Introduction

Recruitment of care home staff to research studies is recognised as challenging. This was further exacerbated by the COVID-19 pandemic and the associated negative media portrayal of care home workers. Social media use has surged since the onset of COVID-19 lockdowns, offering a plausible approach to understanding the barriers to care home research recruitment and gaining insight into public perceptions of care home workers.

### Aim

To utilise comments from two Facebook recruitment posts to: 1) gain an understanding of potential barriers to recruitment of healthcare workers (HCWs) in UK care homes, and 2) explore public sentiment towards care home research and care homes in the context of the COVID-19 pandemic.

### Methods

This cross-sectional study analysed comments from two Facebook posts (available June-October 2021) advertising a separate study on psychological support for care staff during the pandemic. This study was situated within a larger investigation into the mental health and wellbeing of care home staff and employed both qualitative analysis and quantitative methods (word count and correlations between words used and between posts).

### Results

Three themes were identified from the qualitative analysis: support, mistrust and blame. There was a greater use of words associated with support and negative emotive words in

**Funding:** This study was funded by RCN Foundation. The award was received by the lead author. The funder did not play any role in the study design,data collection and analysis, decision to publish, or preparation of the manuscript.

**Competing interests:** The authors have declared that no competing interests exist.

post 2. Post 2 comments featured significantly more choice words and first-person singular pronouns than post 1 which indicated a resentful sentiment from those who advocate freedom of choice and control. Discussion of mistrust towards researchers was most prominent in post 1 indicating the importance of relationship building between researchers and HCWs in UK care homes. With attribution to blame, there was a larger range of negative emotion words than positive emotion words.

## Discussion and conclusion

Taken together our findings offer novel insights into why recruitment to care home research during the pandemic including the use of social media might be problematic. Social media is a useful tool for recruitment but should not be considered as a one-time input. Researchers should pro-actively engage with the study population from the start using co-design with resident and public groups to support recruitment and ensure these populations are accurately represented within research.

## Introduction

Since 2020, the COVID-19 pandemic has impacted the lives of people across the globe. The pandemic presented several challenges for health care workers (HCWs) in nursing and care home settings who experienced significant adjustments to their working environments as well as an increase in residents' mortality [1]. Additionally, HCWs risked their own health and that of their families while caring for residents with COVID-19 [1]. The Health Belief Model (HBM) offers a comprehensive framework to comprehend the responses of HCWs in nursing and care homes amid the COVID-19 pandemic. In this context, HCWs' perception of the threat posed by the virus, both to their physical well-being and mental health, aligns with the core tenets of the HBM. Studies indicate that HCWs perceived a considerable threat, not only to their physical well-being but also to their mental health, as evidenced by high levels of depression and anxiety [2–5]. Additionally, the perceived lack of support from the UK national governments further heightened the sense of vulnerability among HCWs, with a substantial proportion considering leaving their professions due to the inadequacies of support systems [6].

The negative psychological impact on HCWs was exacerbated by stigmatisation, particularly through negative media portrayals of HCWs during COVID-19, by unfairly blaming them for high numbers of COVID-related deaths and imposing restrictions on visits [7]. Here, the HBM helps interpret how these external factors contribute to HCWs' perceptions of susceptibility and severity of the threat posed by the pandemic.

In response to the pandemic and to support all essential workers, Public Health England introduced and recommended a free-to-access online version of Psychological First Aid (PFA) training in June 2020 [8]. The PFA was originally designed as a brief training course to help those involved in disasters and/or traumatic events to reduce their initial distress and enhance their long-term coping [9]. Some evidence suggests that PFA might be useful for HCWs [10]. However, another systematic review of usefulness of PFA for HCWs found that there was a lack of empirical evidence, and many recommendations were based on expert opinions [11]. Taken together, the need to derive tailored evidence-based psychological support to HCWs is pressing. Our study into PFA for HCWs prompted this work due to challenges with recruitment and the perceived possibilities of social media.

Several barriers to research within care homes are well established and have been amplified by the pandemic [12]. Notably, recruitment of care home staff to research studies is recognised as challenging [13]. There could be several reasons for this. Firstly, the impact of COVID-19 likely exacerbated this issue with increased workload, reduction in staff, and subsequent reports of exhaustion among the HCWs. Secondly, the impact of the negative media portrayal of care home staff during the pandemic may have increased their reluctance to participate in research [12]. Thirdly, limited communication pathways to inform care homes about research opportunities can lack clarity. For example, access to work email or work computers by HCWs has been reported as problematic and therefore research information is difficult to distribute [14]. Despite this, it is important for care homes to have opportunities to participate in research, not only as a means to inform future research but as a direct benefit to HCWs by improving their well-being, socialisation and providing a therapeutic benefit for them, as well as opportunities for their voices to be heard [15,16]. Thus, using different approaches and strategies to engage them in research is imperative in addition to some UK initiatives such as ENRICH (ENabling Research In Care Homes) network [17].

Attempts to minimise the spread of COVID-19 resulted in worldwide lockdowns with social media becoming the preferred means of interaction within the public [18]. Hence, the expanding the use of social media provides an alternative approach to attract and recruit research participants. Facebook has the largest social media following with 2.3 billion users worldwide and has been used previously to gain perspectives of hard to reach groups [19–21]. In this paper, we are reporting on the responses to a Facebook recruitment advert for a national cross-sectional survey of UK care home HCWs conducted between June and October 2021 to provide insight into potential barriers to recruitment. Given that the recruitment advert was accessible to the general population, we believed that understanding the public's perception of the proposed research study and more generally care homes during the pandemic, might usefully inform future recruitment from care and nursing homes and psychological research and intervention development for that population. Thus, our aims, informed by the HBM, were to: 1) gain an understanding of potential barriers to recruitment of HCWs in UK care homes, and 2) explore the public sentiment towards care home research and care homes in the context of the COVID-19 pandemic.

## Materials and methods

### Study design

A cross-sectional retrospective observational review of comments made by the public to specific social media recruitment posts (n = 2) was undertaken. This secondary data approach was chosen to capture the public perceptions as it allowed for unfiltered opinions. It also avoided social desirability biases such as the Hawthorne effect where participants may change their behaviour, or censor their opinions, if they are aware they are being studied [22,23]. The data collection and analysis method complied with the terms and conditions of the Facebook data source.

### Data collection

The comments were manually extracted on a Microsoft Excel spreadsheet from two Facebook posts advertising a research opportunity for a study on psychological support for care staff during the pandemic. Data were stored on the Northumbria University secure server and only the research team had access to it. Post 1 (see Fig 1) was a sponsored post from Northumbria University's official Facebook page and set to reach adults in the UK. The recruitment advert in

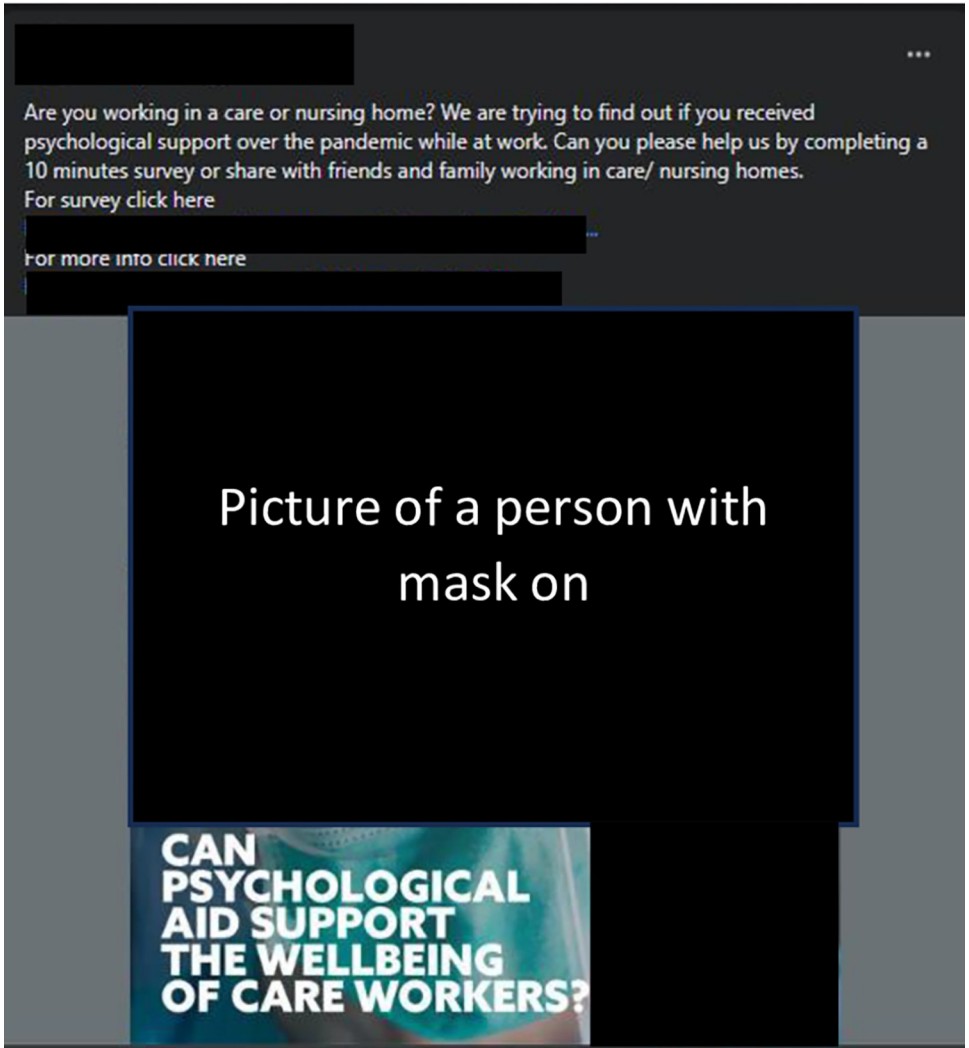

**Fig 1. University sponsored recruitment post.**

this post was hosted on the university's Facebook page and included university branding and reference to the funding organisation. Post 2 (see Fig 2) was advertised from a Facebook page created by the author (MS) but not affiliated with Northumbria University, which advertised research relating to the impact of COVID-19 on mental health and quality of life. Both posts were live at the same time between June and October 2021. MS had author access and therefore an opportunity to reply to questions on the Facebook Post 2, while this was not the case with the Facebook Post 1.

## Ethical considerations

Ethical approval was granted from the Faculty of Health and Life Sciences at Northumbria University REF:32662NU and consent was waived.

## Analysis

Comments were qualitatively analysed using the 6-steps of thematic analysis defined by Braun and Clarke [24]. All members of the team familiarised themselves with the data. During phase

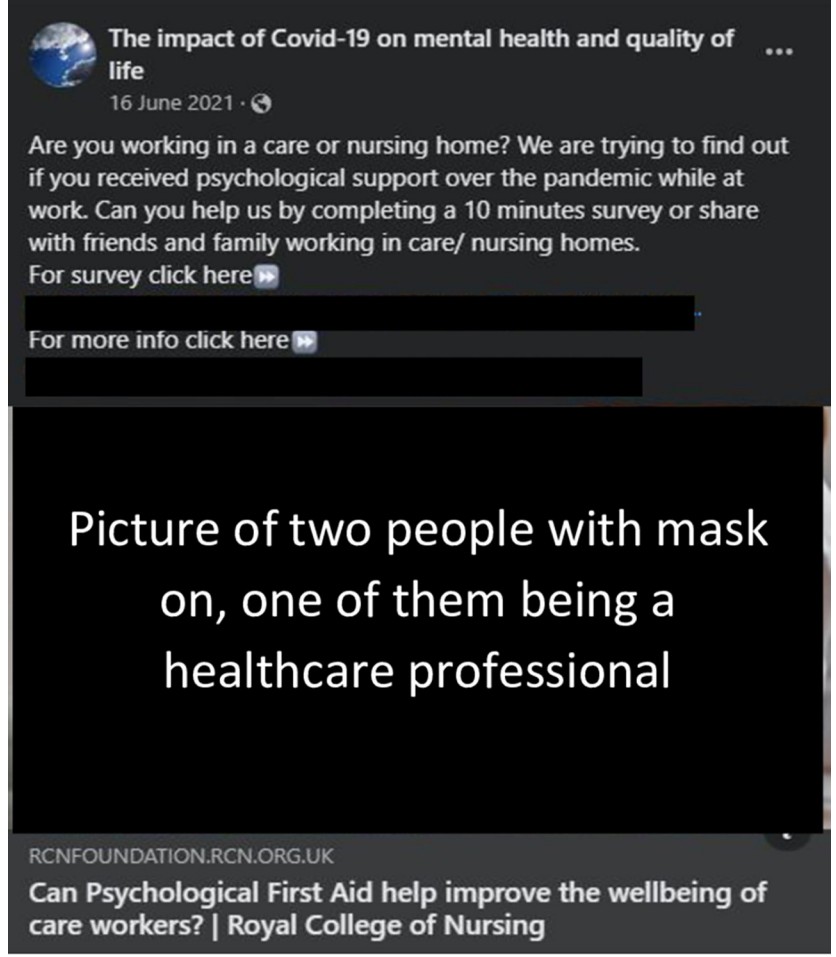

**Fig 2. Facebook group recruitment post.**

two and three, comments were independently coded by MS, CM and MB with initial codes inductively generated and then grouped into themes via collaborative discussion. Subsequently, phase four and five followed, where all the research team met and reviewed if the initial themes were supported by the code extracts and the overall data set. Themes were defined and named with the final names of the key themes and sub-themes agreed by consensus. A report (phase 6) on the findings was agreed by all team members.

The quantitative analysis included exploration of word usage. Several key word categories were identified from the findings of the thematic analysis. Use of these word categories were counted and recorded. Singular and plural pronoun use were quantified to gain insight into perceptions of personal responsibility in the context of the pandemic, displacement of responsibility and locus of control. Table 1 displays word categories and word bank. Finally, the number of words used in each comment was recorded.

A series of t-tests were conducted to compare use of each word category and comment length between posts 1 and 2. Correlations were conducted to explore relationships between the word categories in each post.

**Table 1. Word categories.**

| Word Category | Word bank |
|---|---|
| Positive emotive words | Amazing, funny, glad, happy, loved, loving, respect, safe, trust |
| Negative emotive words | Abuse, burnout, depressing, disgusting, evil, exhausted, fear, hard, horrific, loneliness, negative, overwhelmed, sadly, scary, shame, stress, stressed, stresses, stressing, suffer, suffering, torture, trauma, upsetting, vulnerable, worried |
| First person pronouns (singular) | I, me, my |
| First person pronouns (plural) | We, us, our |
| Second person pronouns | You, your |
| Third person pronouns (singular) | He, him, his, she, her, hers |
| Third person pronouns (plural) | They, their |
| Vaccine | Clotshot, injected, jab, jabs, klotshot, unvaccinated, unvaxxed, vaccinate, vaccinated, vaccine, vaccines, vax, vaxxed |
| Support | Help, support |
| COVID-19 pandemic | Coronavirus, covid, pandemic, |
| Choice | Choice, control, coercing, decision, enforcing, force, forced, forcing |

## Results

199 data points were collected across the two posts, 133 from post 1 and 66 from post 2. These comprised of 90 main comments and linked replies from 94 contributors to post 1, and 26 main comments and linked replies from 38 contributors to post 2. Post 2 reached 14983 people and 20 shares, but we were unable to retrieve this data for post 1.

### Themes

Three themes were identified from the data: i) support, ii) mistrust, and iii) blame.

**I. Support.** Twenty-three data points made reference to support. Most data points (n = 20) relating to this theme highlighted a lack of support for HCWs in care homes and the negative consequences for their well-being as a result. It is a reasonable assumption that if care home staff are physically and mentally exhausted there will be no 'reserve' for them to consider participating in research.

> *"I'm absolutely mentally and physically exhausted, lost a stone in weight with stress, and had no support, myself and staff are done in–Contributor 98 (Post 2)*

> *". . . as for support I get more support of my 6-year-old Primark knickers than I do from the care industry"—Contributor 123 (Post 2)*

Those that received support suggested this was beneficial. Whilst HCWs reported benefits from support, including their mental health and coping with the demands of care home work, the direct connection between support and research recruitment is largely theoretical i.e. more time/headspace they have, the more opportunity they would have to partake in research.

> *"Yes. Co counselling gets rid of stress. Life is then much easier. Can be learned and troubles eased."—Contributor 23 (Post 1)*

> *"Yes we did and it was very beneficial to all our staff"—Contributor 101 (Post 2)*

**II. Mistrust.** Ninety data points were categorised as referencing mistrust, with the majority (n = 55) relating to vaccines, followed by (n = 35) relating to lack of trust towards the government and researchers. Often those working for the Government and researchers were not seen as different by respondents, but rather similarly mistrusted as hierarchies.

Vaccine issues were the most commonly discussed topic within the comments on both posts, with n = 55 data points referencing this subject. Two sub-themes were identified from the data: a) lack of choice, and b) harm.

Participants shared views that emphasised the wariness placed by some about local authorities in relation to COVID-19 and the potential adverse effects and harm of the COVID-19 vaccine which ultimately led to suspicion and mistrust of both government authorities and researchers.

> "*I've chosen vaccine, but it was my choice*"–Contributor 105 (Post 2)

> "*no one should be forced to vaccinate*"—Contributor 105 (Post 2)

Lack of autonomy and coercion were expressed and the consequences of care staff not receiving the vaccination were also discussed in relation to threatened job losses.

> "*I definitely don't want the vaccine but stand to lose a job I've done for 30 years and love*"–Contributor 96 (Post 2)

> "*we're fine—apart from the slight stress caused by being fired because we don't want the vaccine.*"—Contributor 86 (Post 1)

> "*Sure also it will help if the authorities stop blackmailing them No vax No job*—Contributor 10 (Post 1)

There was also mistrust expressed by a few participants in relation to the risks and potential harm associated with receiving the vaccine and concerns around its efficacy. Those respondents seemed to be influenced by personal experience of the COVID-19 vaccine related harm.

> "*my husband's son-in-law ended up with blood clots and a stroke*"–Contributor 18 (Post 1)

> "*hell, it's disgusting my friends that have had the dodgy vaccine are dropping like flies one friend had a stroke the day after. So Evil.*"–Contributor 90 (Post 1)

> "*Do these jabs work? No. Two of my friends had jabs sadly one died of COVID. Other people tell me the same, they had jab one still got COVID. The lady I knew had the two jabs yet still got caught COVID and passed away.*"–Contributor 18 (Post 1)

On occasions the effects of mistrust led to calls for vengeance.

> "*The Government and Media should face a trial over this fake PLANDEMIC. . . .and should face charges of Crimes against Humanity.. then Executed!!!!*"–Contributor 94 (Post 1)

> "*More criminality by government lackies*"–Contributor 43 (Post 1)

Mistrust was also expressed towards the researchers conducting the study, academic institutions and research more widely. This included scepticism towards the evidence and wasteful tax expenditure.

*"Think a bit late Northumbria University. I don't buy your doing this survey to find out if care workers had phycological support. I think the government wants to know what carers are thinking and feeling right now about being blackmailed and victimised to getting vaccinated to keep their jobs."–Contributor 17 (Post 1)*

"Someone somewhere wants to create another ology to make more money out of the taxpayer."–Contributor 20 (Post 1)

*"97% of all scientists agree with the people funding them"–Contributor 26 (Post 1)*

Multiple comments (n = 9) also served to minimise the impact or severity of COVID-19 suggesting a mistrust of the scientific evidence being presented and a disbelief that there was a pandemic

*"How many more masks do you want to wear for a flu with a survival rate of 99.9%?"–Contributor 95 (Post 1)*

"What pandemic?"–Contributor 42 (Post 1)

"There is no pandemic"—Contributor 46 (Post 1)

*"Ask sage they'll know best"–Contributor 16 (Post 1)*

**III. Blame.** Fifteen data points referred to blame. Blame was palpable at multiple levels (care home staff, NHS, government and academia) and, on occasions, were extreme and expressed as seeking vengeance. Further blame comments referred to the role of the government and NHS. The poor relationship between NHS and care home staff was also exemplified.

*"Our NHS colleagues came into our home and called us murderers for losing as many residents, even though it was them that placed an untested patient into the home, because at the time it wasn't government guidelines, and thus caused the virus to spread in our home, a dementia unit with no chance at all of social distancing, isolation nothing, but we get the Flack and criticism and our NHS colleagues get the praise, I'm sorry but local NHS trust look down their noses at care staff in homes"–Contributor 111 (Post 2)*

*"This happened after the government working in tandem with the NHS introduced INHUMANE TORTURE TECHNIQUES."–Contributor 64 (Post 1)*

Blame was also expressed towards managers and the government in relation to lack of adequate funding and resources.

*"The reason the government are increasing the NI contributions is to keep these inept useless managers, team leaders, social workers, managers and their managers in care homes in jobs. They are bleeding the system dry."–Contributor 17 (Post 1)*

*"That is the truth! CEO's and directors will find that a lot of expenditure is spent on middle management, when a better strategy would be to employ competent people who can log directly into a system of work. 70-80k per annum is a lot of money to pay one individual unnecessarily when people/ the workforce have access to use simple technology."–Contributor 62 (Post 1)*

There were a few extreme views of public sentiment seeking vengeance towards care home staff for their role as enabling and allowing residents to be vaccinated.

"*The carers that were complicit to watching residents die of loneliness and watching the results of vaccinating our elderly should have no choice as the residence had no choice. What goes around comes around*".–Contributor 18 (Post 1)

"*I have sympathy for those having to take the jag but if you're a carer who watched what they did to the elderly then yes, it's karma. You all should have stood up when the rest of us were on the streets telling you all you were being conned.*"–Contributor 37 (Post 1)

## Word count analysis

Table 2 displays descriptive statistics and t-test results for the word count analysis.

Comments on post 2 ($M$ = 1.33, $SD$ = 2.14) featured significantly more first-person singular pronouns than the comments on post 1 ($M$ = 0.20, $SD$ = 0.65), $t(197)$ = -5.60, $p < .001$.

Post 2 comments also included significantly more negative emotion words (Post 2 $M$ = 0.45, $SD$ = 0.73, Post 1 $M$ = 0.20, $SD$ = 0.44), $t(197)$ = -3.03, $p < .001$, words associated with support (Post 2 $M$ = 0.21, $SD$ = 0.45, Post 1 $M$ = 0.06, $SD$ = 0.24), $t(197)$ = -3.13, $p < .001$, and words associated with choice (Post 2 $M$ = 0.32, $SD$ = 0.64, Post 1 $M$ = 0.09, $SD$ = 0.34) $t(197)$ = -3.31, $p < .001$, than comments on post 1.

## Correlations

Table 3 displays correlations between word categories in posts 1 and 2.

Use of positive emotion words was positively correlated with use of first-person singular pronouns in both posts (Post 1 $r$ = .19, $p$ = .03, Post 2 $r$ = .31, $p$ = .01), first-person plural pronouns in post 2 ($r$ = .31, $p$ = .01), and second-person pronouns in post 1 ($r$ = .22, $p < .001$).

Use of negative emotion words was positively correlated with use of first-person singular pronouns ($r$ = .36, $p < .001$), first-person plural pronouns ($r$ = .35, $p < .001$), and third-person plural pronouns ($r$ = .36, $p < .001$) in post 2. Negative emotion word use also positively correlated as follows: with words associated with support ($r$ = .27, $p$ = .03); the COVID-19 pandemic ($r$ = .38, $p < .001$) in post 2; and words associated with vaccines in post 1 ($r$ = .18, $p$ = .04).

**Table 2. T-test analysis for word counts.**

|  | Post 1 | | | | Post 2 | | | | |
| --- | --- | --- | --- | --- | --- | --- | --- | --- | --- |
|  | **Mean** | **SD** | **Max** | **Min** | **Mean** | **SD** | **Max** | **Min** | ***t*** |
| Positive emotion words | 0.09 | 0.34 | 2 | 0 | 0.12 | 0.33 | 1 | 0 | -0.62 |
| Negative emotion words | 0.20 | 0.44 | 2 | 0 | 0.45 | 0.73 | 3 | 0 | -3.03** |
| First person singular pronouns | 0.20 | 0.65 | 5 | 0 | 1.33 | 2.14 | 11 | 0 | -5.60** |
| First person plural pronouns | 0.26 | 0.69 | 5 | 0 | 0.53 | 1.33 | 7 | 0 | -1.92 |
| Second person pronouns | 0.34 | 1.11 | 10 | 0 | 0.23 | 0.58 | 3 | 0 | 0.76 |
| Third person singular pronouns | 0.05 | 0.26 | 2 | 0 | 0.21 | 0.90 | 6 | 0 | -1.89 |
| Third person plural pronouns | 0.38 | 1.02 | 8 | 0 | 0.58 | 0.86 | 3 | 0 | -1.32 |
| Vaccine | 0.32 | 0.63 | 4 | 0 | 0.39 | 0.74 | 4 | 0 | -0.77 |
| Support | 0.06 | 0.24 | 1 | 0 | 0.21 | 0.45 | 2 | 0 | -3.13** |
| Choice | 0.09 | 0.34 | 2 | 0 | 0.32 | 0.64 | 3 | 0 | -3.31** |
| C-19 pandemic | 0.18 | 0.47 | 3 | 0 | 0.50 | 1.27 | 9 | 0 | -2.57* |
| Word count | 24.12 | 57.66 | 641 | 0 | 31.61 | 32.60 | 151 | 0 | -0.98 |

*$p < .05$

**$p < .001$.

**Table 3. Word category correlations.**

| | | Negative emotion words | First person singular pronouns | First person plural pronouns | Second person pronouns | Third person singular pronouns | Third person plural pronouns | Vaccine | Support | Choice | C-19 pandemic |
|---|---|---|---|---|---|---|---|---|---|---|---|
| | | r | r | r | r | r | r | r | r | r | r |
| Positive emotion words | Post 1 | -.02 | .19* | .00 | .22** | .03 | .10 | -.06 | .03 | -.01 | -.01 |
| | Post 2 | .28* | .31* | .31* | .10 | -.04 | .18 | .31* | .14 | .11 | .04 |
| Negative emotion words | Post 1 | | .09 | .08 | -.10 | .11 | .04 | .18* | .10 | .13 | .04 |
| | Post 2 | | .36** | .35** | .01 | -.15 | .36** | -.05 | .27* | -.02 | .38** |
| First person singular pronouns | Post 1 | | | -.07 | .55** | .07 | .13 | .32** | .02 | .12 | .18* |
| | Post 2 | | | .22 | .11 | .13 | .00 | .44** | .29* | .26* | .08 |
| First person plural pronouns | Post 1 | | | | .22* | .39** | .35** | .02 | .14 | .03 | .00 |
| | Post 2 | | | | .24* | -.08 | .28* | .33** | .04 | -.04 | .08 |
| Second person pronouns | Post 1 | | | | | .20* | .40** | .07 | .01 | .14 | -.09 |
| | Post 2 | | | | | -.06 | .32** | .47** | .05 | .30* | -.05 |
| Third person singular pronouns | Post 1 | | | | | | .50** | .04 | -.05 | -.06 | -.08 |
| | Post 2 | | | | | | -.14 | .17 | .00 | -.01 | -.05 |
| Third person plural pronouns | Post 1 | | | | | | | .06 | .03 | -.06 | -.13 |
| | Post 2 | | | | | | | -.05 | .12 | .00 | -.03 |
| Vaccine | Post 1 | | | | | | | | .02 | .15 | .44** |
| | Post 2 | | | | | | | | -.16 | .48** | -.05 |
| Support | Post 1 | | | | | | | | | .03 | -.10 |
| | Post 2 | | | | | | | | | -.13 | .08 |
| Choice | Post 1 | | | | | | | | | | .93 |
| | Post 2 | | | | | | | | | | -.14 |

*$p < .05$

**$p < .001$.

Use of words associated with vaccines positively correlated with first-person singular pronouns in both posts (post 1 $r = .32$, $p < .001$, post 2 $r = .44$, $p < .001$), and first-person plural pronouns ($r = .33$, $p < .001$), and second-person pronouns ($r = .47$, $p = .03$) in post 2. Vaccine words also positively correlated with words associated with choice in post 2 ($r = .48$, $p < .001$) and the COVID-19 pandemic in post 1 ($r = .44$, $p < .001$).

Words associated with support positively correlated with first-person singular pronouns ($r = .29$, $p = .02$) in post 2. Choice words were positively correlated with first-person singular pronouns ($r = .26$, $p = .04$) and second-person pronouns ($r = .30$, $p = .01$) in post 2. Words associated with the COVID-19 pandemic were also positively correlated with first-person singular pronouns ($r = .18$, $p = .04$) in post 1.

## Discussion

This paper aimed to gain understanding of potential barriers to recruitment of HCWs in UK care homes and explore public sentiments towards research in care homes during the Covid-19 pandemic by analysing comments on social media recruitment posts for a study of HCWs' well-being. While our analysis primarily focused on themes of support, mistrust, and blame, integrating insights from the Health Belief Model (HBM) enriches our understanding of the underlying mechanisms driving these sentiments. There was a greater use of words associated with support and negative emotive words in post 2. Post 2 comments featured significantly more choice words and first-person singular pronouns than post 1. Discussion of mistrust towards researchers was most prominent in Facebook post 1. With attribution to blame, there was a larger range of negative emotion words than positive emotion words. While the recruitment strategy using social media for the wider study was successful, it was not without challenges. Thus, taken together, our findings offer some insights into why recruitment to care home studies during the pandemic, using social media may have been challenging based on the sentiments shared both by HCWs and by the public.

### Support

Our findings underscore the crucial role of support, encompassing both psychological and logistical assistance, in mitigating the adverse effects of the pandemic on care home staff's well-being. The Health Belief Model provides a valuable framework for interpreting these findings, particularly regarding perceptions of susceptibility, severity, and perceived benefits of seeking support. Perceived susceptibility to negative outcomes, such as mental health challenges, may motivate individuals to seek support services. However, our study suggests that care home staff received little psychological support during the pandemic, aligning with previous research demonstrating low uptake of well-being interventions among HCWs [11,25]. This perceived lack of support may exacerbate feelings of vulnerability and contribute to increased depression, anxiety, and insomnia [26–28]. The positive relationship between words associated with support and negative emotive words in our word count analysis further underscores the importance of addressing support deficits to alleviate psychological distress among care home staff. Clearly, care home staff's physiological and safety needs would require to be met before they could participate in research. Meeting these fundamental needs are important considerations when recruiting to care home studies. Researchers must ensure physical comfort is obtained from adequate breaks, minimal participant burden and creating an environment for psychological safety i.e., confidentiality etc. Importantly, ensuring care home staff have an opportunity to participate in research has important ethical principle i.e. fairness and equity of opportunity.

### Mistrust

Mistrust was a prevalent theme throughout the data, particularly in relation to the effectiveness and risks of the COVID-19 vaccine, authorities, and researchers. Public and care home staff mistrust is important as it can affect all levels of trust in societal and social systems [29]. For example, mistrust in authorities can also become mistrust in researchers. Trust depends on people's previous experiences, with negative experiences being more powerful at building mistrust, than positive experiences building trust [30]. Therefore, resources need to be directed towards creating lots of positive experiences with care home workers to build trust, and subsequently increase their likelihood of engaging with research projects. Additionally, how trust is perceived is dependent upon individuals' perceptions of what is fair [31]. Many respondents in this study felt that it was unfair to enforce vaccinations for care home staff, as this removed

individual autonomy. Some posts even suggested that care home staff should be vaccinated in retribution for allowing elderly residents to be vaccinated. To understand care home workers values there is a need for researchers to build relationships with them.

Additionally, there were intense views around the efficacy and safety of the COVID-19 vaccine with some reporting serious adverse medication events from people known to them. Of course, these could be extreme views by those who have had recent negative experiences, otherwise known as proximity fallacy. These posts contained a discussion about vaccine and related choices and consequences, despite no mention of vaccine in the recruitment post. However, the timing of the recruitment post coincided with public health and government discussions and activities around the vaccine in general and the compulsory vaccination for HCWs, placing the responsibility to protect vulnerable people (care home community) on HCWs (individual level). Thus, it is not surprising to see vaccine and choice discussion on the posts.

Post 2 comments featured significantly more choice words and first-person singular pronouns than post 1. In addition, post 2 had stronger associations between vaccine words and choice words or positive emotion or singular pronouns in comparison to post 1. It is possible to assume that the people commenting on post 2 felt they had more choice or more choice should be available to people regarding the vaccine, but also could indicate a higher internal locus of control where they have a personal responsibility around limiting the transmission of the disease. This fits with the theory of locus of control which looks at the extent to which people believe they control their lives (internal locus) or if others control their life, such as the influence and power of government or other forces (external locus) [32]. This could be particularly important in the context of the global pandemic and the mental health of the public and HCWs. Having a higher external locus can exacerbate negative feelings, feelings of threat and having depressive thoughts in comparison to having a higher internal locus where people feel more in control and that their actions matter [33,34]. What this means for barriers to recruitment is that those that feel they have less control over their life are more likely to have mistrust towards government and researchers and are therefore less likely to participate in research. Thus, as mentioned before, doing work that can increase trust in researchers by building relationships with care homes, can change the perception HCWs have towards researchers.

Findings relating to mistrust of researchers and perceptions of government sponsorship highlight the need for better collaboration and sharing of power between researchers and HCWs. All stages of the research process would benefit from co-design, particularly preparation of recruitment and data collection materials to ensure they appeal to and are suitable for the population. Researchers should be aware of the potential impact of including institutional and funders' information on recruitment posts. Discussion of mistrust towards researchers was most prominent on post Facebook 1. Given the clear need for more research to be conducted with HCWs, it is important that as researchers we work to foster relationships with this population–research champions to promote collaboration with care and nursing homes. This also highlights the importance of researcher responses to comments–where recruitment posts attract some negative comments this may 'taint' the post and put off people who might otherwise have considered taking part. Researchers engaging with the comments to provide and clarify information about the study (as would happen face to face) might help to limit this and increase the credibility of the post.

## Blame

Finally, a preponderance of negative emotion words was apparent and attribution of blame in relation to care home staff was evident within our findings. Applying the Health Belief Model,

individuals' perceptions of blame may reflect their beliefs about the causes and controllability of adverse outcomes. There was a larger range of negative emotion words than positive emotion words. Negative emotion words were more likely to be higher valency e.g., depressed, trauma, suffering, than the positive emotion words which were typically lower valence e.g., funny, glad, happy. This suggests negative emotions experienced and expressed were more intense and warrants the need for development and accessibility of public health interventions that can address these intense negative emotions. Use of negative emotions words in the comments on post 1 were typically used in reference to others e.g., care home residents being lonely, whereas in post 2 they were more often used in reference to care home staff being burnt out and exhausted. This further demonstrates the dominant external locus of control among the participants, but also highlights the general worry about the vulnerable and the underappreciated members of the public and dissatisfaction about how the challenges of the pandemic were handled.

## Strengths and limitations

It is important to acknowledge the strengths and limitations of using social media such as Facebook data as a research source. One of the strengths is that is a quick and easy way of accessing public's views on specific topics without needing an explicit consent since the information is in public domain [35]. Further, as comments were generated naturalistically, we can assume that opinions expressed are uncensored and not subject to the demand characteristics and social desirability bias often observed within the context of experimental designs. However, this method of data collection also presents notable limitations. Firstly, demographic information could not be collected for those who contributed to the comments. It was unclear from some posts whether these were a lay public perspective or specifically experiences of care home workers. While it can be inferred that contributors to the comments on post 2 were more likely to be HCWs due to the content of the comments and the target audience of the page, similar inferences cannot be made for post 1. Secondly, difficulties arose with determining which were genuine opinions and experiences and which comments were posted to be deliberately controversial or confrontational (particularly in the context of a sensitive and potentially divisive topic such as the COVID-19 pandemic). As such, all comments were extracted and treated as valid data points, however this may mean that excessively negative, anti-vaccine or anti-pandemic opinions were over-represented in the data.

Also, word count analysis does not consider the semantics of the word in the context of the sentence [36]. Many of the positive emotive words were used to convey sarcasm or irony. Contributors to social media discussions may represent a self-selected group with particular views, potentially skewing the data towards certain sentiments and limiting the generalisability of the findings [37]. Finally, we want to acknowledge that the method used is insufficient to address aim 1 entirely and future qualitative studies should be undertaken to explore this further.

The negative media portrayal of care homes during the COVID-19 pandemic can be attributed to several interconnected factors [38]. Heightened mortality rates within care homes, often resulting from the virulent nature of the virus, have become focal points in media coverage, creating a perception of inadequacy in crisis management. Reports on insufficient protective measures, including shortages of essential equipment, and staffing challenges, such as shortages and overwork, contribute to the negative image [39]. Communication issues, both within care homes and between stakeholders, are highlighted, further impacting the perception of care home efficacy. Public and governmental responses, including policies and financial support, are scrutinized, adding another layer to the narrative. Stigmatization of care homes as COVID-19 hotspots and potential sensationalism by the media contribute to a nuanced and

often negative portrayal. This complex interplay of factors underscores the multifaceted nature of the media's influence on public perception during the pandemic.

## Conclusion

Social media is a useful tool for recruitment but should not be considered as a one–off strategy for recruitment but rather built in as an integral avenue for recruitment to research with appropriate ethics approval especially with hard-to-reach groups. Researchers should aim to engage with their target populations to provide as much information about the work as possible and help address misconceptions in the comments which may 'taint' the post. Also, useful for researchers working with the HCW population, is to engage in co-design from the start and include PPI groups in recruitment. The implementation of research champions in these settings would benefit care home staff to be proactive in building relationships with researchers and aid recruitment to ensure these populations are properly represented within research.

## Author Contributions

**Conceptualization:** Mariyana Schoultz.

**Data curation:** Mariyana Schoultz, Claire Mcgrogan.

**Formal analysis:** Mariyana Schoultz, Claire Mcgrogan, Clare Carolan, Leah Macaden, Michelle Beattie.

**Funding acquisition:** Mariyana Schoultz.

**Investigation:** Mariyana Schoultz.

**Methodology:** Claire Mcgrogan.

**Validation:** Leah Macaden.

**Writing – original draft:** Mariyana Schoultz, Claire Mcgrogan, Clare Carolan, Michelle Beattie.

**Writing – review & editing:** Clare Carolan, Leah Macaden, Michelle Beattie.

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
