## [Decision Letter · Decision Letter 0]

12 Oct 2022

PONE-D-22-20731Exploring barriers to recruitment for a care home study during the COVID-19 pandemic: The influence of social media recruitment posts and public sentiment.PLOS ONE

Dear Dr Schoultz,

Thank you for submitting your manuscript to PLOS ONE. After careful consideration, we feel that it has merit but does not fully meet PLOS ONE’s publication criteria as it currently stands. Therefore, we invite you to submit a revised version of the manuscript that addresses the points raised during the review process. Please kindly go through all the comments made by the reviewers and kindly make point-by-point response.

We look forward to receiving your revised manuscript.

Kind regards,

Nusrat Homaira

Academic Editor

PLOS ONE

Journal Requirements:

2. In your Methods section, please include additional information about your dataset and ensure that you have included a statement specifying whether the collection and analysis method complied with the terms and conditions for the source of the data.

“We would like to thank the funder RCN Foundation for making this research possible.”

“This study was funded by RCN Foundation.

The award was received by the lead author.

The funder did not play any role in the study design,data collection and analysis, decision to publish, or preparation of the manuscript.”

“None”

Reviewers' comments:

Reviewer's Responses to Questions

**Comments to the Author**

1. Is the manuscript technically sound, and do the data support the conclusions?

Reviewer #1: Partly

Reviewer #2: Yes

2. Has the statistical analysis been performed appropriately and rigorously? 

Reviewer #1: Yes

Reviewer #2: Yes

3. Have the authors made all data underlying the findings in their manuscript fully available?

Reviewer #1: Yes

Reviewer #2: Yes

4. Is the manuscript presented in an intelligible fashion and written in standard English?

Reviewer #1: Yes

Reviewer #2: Yes

5. Review Comments to the Author

Reviewer #1: The manuscript, “Exploring barriers to recruitment for a care home study during the COVID-19

pandemic: The influence of social media recruitment posts and public sentiment” describes the factors influencing recruitment for care homes. The paper can be considered for publication, but I also think the article needs major revisions.

In the abstract, the rationale for this study should be strengthened along with what additional knowledge this paper will add to the existing literature. The methods are sufficient to address the 1st objective: gain an understanding of potential barriers to recruitment of HCWs in UK care homes

Specific comments

Abstract:

Page 2, lines 21-22 “Recruitment of care home staff to research studies is recognised as challenging, which got further exacerbated by the pandemic”. Please be specific here “COVID-19 Pandemic”.

The authors need a sentence to rationalise the study. What additional knowledge this study will add to the existing literature can be highlighted.

If space allows, briefly discuss the two Facebook posts here.

Under methods, the timeline of review/duration of study is required.

Manuscript:

Introduction

Page 5, lines 85-86, “- For example, HCWs often don’t have access to work email or work 86 computer and therefore research information is difficult to distribute.” -need a reference to support this statement

AIMS: Aims could be merged with the introduction. It can be the last sentence of the final paragraph under the introduction. If possible, combine the two aims into one.

Methods:

The timeline of study is missing here.

The authors need to mention the study site/geographical area of the study.

The authors need to introduce the data collection team here, along with their background. How did they collect the data?

How was the data extracted? Did the team use any validated form of data extraction?

The authors also need to clarify how the data were stored and managed. What technology was used to extract data (Manually, AI, etc.)

Did the study receive any ethics approval? If yes, you need to mention that at the end of the method section. If not, the authors need to clarify , why.

Page 7, line 137 “objectives 1 and 2 were” - There is a difference between the objectives and aims. It would help if you were consistent. How were the codes developed? Did you use any code definitions? Was there any discrepancy among the coders? If yes, how were they minised?

Did you collect data on Emoji?

Results:

Page 9, line 157, “199 data points”—What do you mean by that? What about like , share and Emoji? Are they only comments?

Anyone can comment on the facebook post. How many of the people who commented were HCWs? How did you differentiate HCWs from the general public?

Discussion

The first paragraph of the discussion section should summarise the findings that address the two study objectives. In the limitation section, the author may acknowledge that the method is insufficient to address objective 1 entirely and recommend future qualitative studies. However, the importance of the data generated from this study should be highlighted. The future use of study findings needs to be discussed.

Page 17, line 353 – “the recruitment advert 354 in this post was hosted on the university’s Facebook page and included university 355 branding and reference to the funding organization”. Seems more like a methodology section.

Reviewer #2: Thank-you for the opportunity to review this manuscript. This research is extremely relevant in current global context- exploring public perceptions to recruitment posts (n=2) on Facebook during the COVID-19 pandemic. Overall the paper is well-written and has great "real-world" relevance. The authors succinctly demonstrate the prevalence of the HCW in UK care homes cohort.

- Background: Succinct background section with appropriate introduction to the topic.

- Materials and methods:

* An ethics statement is provided in the submission checklist but not within the body of the manuscript. This should be added. I assume a "waiver of consent" was provided? Were participants informed that their posts would be analysed after the fact e.g. a comment on the post advising that their comments would be analysed?

* I appreciate the authors use of both qualitative and quantitative analysis- very interesting!

-Results

* Are the authors able to provide any metrics relating to "reach" of the two Facebook posts? They provide figures around number of data points and number of data points. Can they provide further details around potential reach? e.g. how many followers for each page, how many views each post received, reposts etc.

*Both social media posts contained a link to a survey. How did the comments/results analysed in this manuscript compare to the findings of the survey study?

Ethical consideration: have the two Facebook posts since been removed from public view? Once published in the journal, it is possible that a PLOS ONE reader could locate the two posts on Facebook and subsequently identify participants in this study. I appreciate that Facebook users are publicly posting on the two posts and thus should be aware that their comments can be viewed by members of the public. However, all reasonable attempts should be made to deidentify research participants especially considering that they did not provide informed consent for this study.

6. PLOS authors have the option to publish the peer review history of their article (what does this mean?). If published, this will include your full peer review and any attached files.

Reviewer #1: No

Reviewer #2: No

---

## [Author Response · Author response to Decision Letter 0]

22 Dec 2022

Dear Editors and Reviewers 

Thank you for your time and suggestions on how to improve the manuscript titled ‘Exploring barriers to recruitment for a care home study during the COVID-19 pandemic: The influence of social media recruitment posts and public sentiment.’

We have addressed all Editors and Reviewers comments and we think the changes significantly improved the manuscript. We have provided responses in the table below to each individual suggestion. Two corrected versions of the manuscript have been submitted. One with highlighted changes to show where the changes were made and one clean version as requested.

We have also adhered to the Journal Requirements below:

Answer. Formating corrected

2. In your Methods section, please include additional information about your dataset and ensure that you have included a statement specifying whether the collection and analysis method complied with the terms and conditions for the source of the data.

Answer: Collection and analysis method complied with the terms and conditions for the source of the data

“We would like to thank the funder RCN Foundation for making this research possible.”

Answer: Acknowledgment statement has been removed and replaced with None

“This study was funded by RCN Foundation.

The award was received by the lead author.

The funder did not play any role in the study design,data collection and analysis, decision to publish, or preparation of the manuscript.”

Answer: The amended statement should state: ”This study was funded by RCN Foundation.The funder did not play any role in the study design,data collection and analysis, decision to publish, or preparation of the manuscript.”

“None”

Answer: "The authors have declared that no competing interests exist.",

Answer: We have made the data available here: https://osf.io/f7sae/

Answer: no restrictions, we have made the anonymous data available

Answer: Data is available here: https://osf.io/f7sae/

Answer: this information was added to the methods section Ethical approval was granted from the Faculty of Health and Life Sciences at Northumbria University REF:32662NU and consent was waived.

Reviewer 1 

In the abstract, the rationale for this study should be strengthened along with what additional knowledge this paper will add to the existing literature. The methods are sufficient to address the 1st objective: gain an understanding of potential barriers to recruitment of HCWs in UK care homes 

A: Thank you for your comment. 

To strengthen the rationale in the abstract, we rewrote the paragraph:

Recruitment of care home staff to research studies is recognised as challenging. This got further exacerbated by the COVID-19 pandemic and the negative media portrayal during the pandemic. With social media becoming the preferred interaction means during the pandemic it became a suitable approach to understand the barriers to recruitment and gain insight into public perceptions of care home workers.

The discussion and conclusion section highlights the additional (novel) knowledge in the area: Taken together our findings offer novel insights into why recruitment to care home studies during the pandemic including the use of social media might be problematic.

Specific comments

Abstract:

Page 2, lines 21-22 “Recruitment of care home staff to research studies is recognised as challenging, which got further exacerbated by the pandemic”. Please be specific here “COVID-19 Pandemic”.

The authors need a sentence to rationalise the study. What additional knowledge this study will add to the existing literature can be highlighted.

If space allows, briefly discuss the two Facebook posts here.

Under methods, the timeline of review/duration of study is required. 

A: Thank you for your comments. We made the following changes on p2 and p3: 

We added the term COVID-19.

To strengthen the rationale in the abstract, we rewrote the paragraph:

Recruitment of care home staff to research studies is recognised as challenging. This got further exacerbated by the COVID-19 pandemic and the negative media portrayal during the pandemic. With social media becoming the preferred interaction means during the pandemic it became a suitable approach to understand the barriers to recruitment and gain insight into public perceptions of care home workers.

The discussion and conclusion section highlights the additional (novel) knowledge in the area: Taken together our findings offer novel insights into why recruitment to care home studies during the pandemic including the use of social media might be problematic.

There was no additional space in the wordcount of the abstract to further discuss the posts here, but this is the information in the methods: Comments were manually extracted from two Facebook posts advertising a study of psychological support for care staff during the pandemic. Comments were analysed qualitatively (thematic analysis) and quantitatively (word count and correlations between words used and between posts). 

We added the timeline for the study in the methods: Facebook posts (available between June-October 2021).

Introduction

Page 5, lines 85-86, “- For example, HCWs often don’t have access to work email or work 86 computer and therefore research information is difficult to distribute.” -need a reference to support this statement

AIMS: Aims could be merged with the introduction. It can be the last sentence of the final paragraph under the introduction. If possible, combine the two aims into one. 

A: Thank you for your comment.

We have added now a reference to support this statement on p5. 

Beattie, M., Carolan, C., Macaden, L., Maciver, A., Dingwall, L., Macgilleeathain, R., & Schoultz, M. (2022). Care home workers experiences of stress and coping during COVID‐19 pandemic: A mixed methods study. Nursing Open.

On p6 we merged the aims with the background section, but we kept the two separate aims for clarity as they both have a slightly different focus.

Methods:

The timeline of study is missing here.

The authors need to mention the study site/geographical area of the study.

The authors need to introduce the data collection team here, along with their background. How did they collect the data?

How was the data extracted? Did the team use any validated form of data extraction?

The authors also need to clarify how the data were stored and managed. What technology was used to extract data (Manually, AI, etc.)

A: Thank you for your comment. The following changes were made on p6

Timeline has been highlighted in data collection section: Both posts were live at the same time between June and October 2021. 

Geographical data has been highlighted in the same section: Facebook page and set to reach adults in the UK.

Information about data collection and storage has been highlighted in the data collection section p6: Comments were manually extracted on an Microsoft Excel spreadsheet (by CM) from two Facebook posts advertising a study of psychological support for care staff during the pandemic. Data was stored on Northumbria University secure server and only the research team had access to it.

Did the study receive any ethics approval? If yes, you need to mention that at the end of the method section. If not, the authors need to clarify , why. 

A: Thank you for your comment. 

We have added a section on ethical considerations: Ethical approval was granted from the Faculty of Health and Life Sciences at Northumbria University REF:32662NU and consent was waived.

Page 7, line 137 “objectives 1 and 2 were” - There is a difference between the objectives and aims. It would help if you were consistent. How were the codes developed? Did you use any code definitions? Was there any discrepancy among the coders? If yes, how were they minimised?

Did you collect data on Emoji?

A: Thank you for your comment.

On p7 e changed objectives to aim for consistency. 

Codes were inductively generated with no predefined coding applied. Data were initially independently coded by three researchers with high levels of agreement of initial code generation. Themes were generated via collaborative consensus by the whole team. We have amended the methods section on p7 accordingly.

We did collect data on emoji, however there wasn’t sufficient data to draw meaningful conclusions from it.

Results:

Page 9, line 157, “199 data points”—What do you mean by that? What about like , share and Emoji? Are they only comments?

A: Thank you for your comment. 

Data points is a wide term and comprises comments and replies. The following has been highlighted on page 9 in the results: 199 data points were collected across the two posts, 133 from post 1 and 66 from post 2. These comprised 90 main comments and linked replies from 94 contributors to post 1, and 26 main comments and linked replies from 38 contributors to post 2. There were 20 shares on post 2, but we didn’t get information on how many shares on post 1. We didn’t consider emoji for analysis as there was insufficient data.

Anyone can comment on the Facebook post. How many of the people who commented were HCWs? How did you differentiate HCWs from the general public?

A: Thank you for your comment. 

We agree that anyone can comment on the post and therefore we highlighted this in the limitation section: 

Limitations: Firstly, demographic information could not be collected for those who contributed to the comments. This means controlling for or comparing between variables such as job title (e.g.,HCW v non-HCW), and country (restrictions varied between countries in the UK at different times) was not possible. It is inferred that contributors to the comments on post 2 were more likely to be HCWs due to the content of the comments and the target audience of the page, however similar inferences cannot be made for post 1.

Discussion

The first paragraph of the discussion section should summarise the findings that address the two study objectives. In the limitation section, the author may acknowledge that the method is insufficient to address objective 1 entirely and recommend future qualitative studies. However, the importance of the data generated from this study should be highlighted. The future use of study findings needs to be discussed. 

A: Thank you for your comment. 

We have summarised the results in the first paragraph of the discussion p15:

Four themes emerged from the data namely support, vaccine, mistrust, and blame. There was a greater use of words associated with support and negative emotive words in post 2. Post 2 comments featured significantly more choice words and first-person singular pronouns than post 1. Discussion of mistrust towards researchers was most prominent in post 1. With attribution to blame, there was a larger range of negative emotion words than positive emotion words. 

We have highlighted the following in the limitation section: Finally, we want to acknowledge that the method used is insufficient to address aim 1 entirely and future qualitative studies should be undertaken to explore this further.

Page 17, line 353 – “the recruitment advert in this post was hosted on the university’s Facebook page and included university branding and reference to the funding organization”. Seems more like a methodology section.

A: Thank you for your comment. 

We have now moved this sentence in the methods section on p6 under data collection and deleted from page 17.

Reviewer 2 Thank you for your comment. 

Background: Succinct background section with appropriate introduction to the topic.

 Thank you for your comment. We appreciate your feedback.

- Materials and methods:

* An ethics statement is provided in the submission checklist but not within the body of the manuscript. This should be added. I assume a "waiver of consent" was provided? Were participants informed that their posts would be analysed after the fact e.g. a comment on the post advising that their comments would be analysed?

* I appreciate the authors use of both qualitative and quantitative analysis- very interesting!

A: Thank you for your comment. 

We have added a section on ethical considerations: Ethical approval was granted from the Faculty of Health and Life Sciences at Northumbria University REF:32662NU and consent was waived.

Thank you for your comment on the analysis, we appreciate it.

-Results

* Are the authors able to provide any metrics relating to "reach" of the two Facebook posts? They provide figures around number of data points and number of data points. Can they provide further details around potential reach? e.g. how many followers for each page, how many views each post received, reposts etc.

*Both social media posts contained a link to a survey. How did the comments/results analysed in this manuscript compare to the findings of the survey study? 

A: Thank you for your comment. 

We provided metrics on reach for post 2, but we couldn’t ret

---

## [Decision Letter · Decision Letter 1]

17 Apr 2023

PONE-D-22-20731R1Exploring barriers to recruitment for a care home study during the COVID-19 pandemic: The influence of social media recruitment posts and public sentiment.PLOS ONE

Dear Dr. Schoultz,

Thank you for submitting your manuscript to PLOS ONE. After careful consideration, we feel that it has merit but does not fully meet PLOS ONE’s publication criteria as it currently stands. Therefore, we invite you to submit a revised version of the manuscript that addresses the points raised during the review process.

We look forward to receiving your revised manuscript.

Kind regards,

Ali B. Mahmoud, Ph.D.

Academic Editor

PLOS ONE

Reviewers' comments:

Reviewer's Responses to Questions

**Comments to the Author**

1. If the authors have adequately addressed your comments raised in a previous round of review and you feel that this manuscript is now acceptable for publication, you may indicate that here to bypass the “Comments to the Author” section, enter your conflict of interest statement in the “Confidential to Editor” section, and submit your "Accept" recommendation.

Reviewer #3: (No Response)

Reviewer #4: (No Response)

2. Is the manuscript technically sound, and do the data support the conclusions?

Reviewer #3: No

Reviewer #4: Yes

3. Has the statistical analysis been performed appropriately and rigorously? 

Reviewer #3: I Don't Know

Reviewer #4: Yes

4. Have the authors made all data underlying the findings in their manuscript fully available?

Reviewer #3: No

Reviewer #4: Yes

5. Is the manuscript presented in an intelligible fashion and written in standard English?

Reviewer #3: No

Reviewer #4: Yes

6. Review Comments to the Author

Reviewer #3: Abstract

The title and bits of the abstract make it sound as if the care home itself (including residents) are the focus of the research (and the research that this sub-study is situated in)

The paper itself is more about public sentiment to care home staff.

I became confused about the post 1 and post 2, and initially thought it was related to the two aims.

Not clear who the participants are in this study.

Background –

It becomes clear that this study is situated in a wider study on the mental health and wellbeing of care home staff. This revelation was a bit late as I initially thought it was about recruiting care homes to research (for residents)

Page 5, line 87, you talk about the fact that its difficult to recruit care homes to research. This is true, but there are some good initiatives out there such as the ENRICH network that are worth mentioning. You do talk about this again in your conclusion.

Methods – I’m not convinced that you have really done a thematic analysis as considered by Braun and Clark. There are about 200 data items in total. This sounds more like a content analysis than a thematic analysis. Have a look at later writing eg Braun and Clarke 2021. As B&C would say, ‘themes don’t emerge’

I think the findings/themes developed are interesting but in order to address what it says in the title, but you need to discuss how these themes act as barriers to recruitment more.

In the methods it’s still not clear who the participants are. By the results this has become clearer that it is likely primarily to be members of the UK public (particularly to post 1). I don’t think this is tidy enough.

Don’t you need ethical approval for this kind of research? Or at least some discussion of ethical issues? See this for example. https://www.gla.ac.uk/media/Media_487729_smxx.pdf

I get the sense that this paper was based on getting some interesting reflections back from your recruitment method and thus a bit of an additional publication.

Results – I thought there was some interesting data in here. The reflections on mistrust of researchers and timing with the required vaccination of English (not Welsh, not sure about Scotland) care home staff to have a COVID-19 vaccination are interesting.

The quantitative analysis seems less useful. I’m not sure what the point of comparing +ve and -ve words for post 1 and 2 are, if you don’t know who your respondents are.

I’m not sure the word count analysis aspect added very much to the paper because there is no way to confidently attribute the number of times a word is said with whether it presents as a barrier to recruitment – but this is mentioned in their limitations.

Discussion and conclusion

I do agree that we need better collaboration between care home staff and researchers.

We still don’t know what the results from the survey reported. Was this method of recruitment ultimately successful or not. Your last sentence of the conclusion suggest it was, but we have no evidence of that.

You reflect on the lack of knowledge of who is responding to the post, but I think this is a more fundamental problem. Those people who reply to facebook adverts with text rather than just a click to ‘find out more’ are more likely to be confrontational, and you know nothing about their background.

Do we know if facebook advertising is more successful at recruiting care home staff than any other section of the society?

Overall style

There are a number of lengthy sentences that could be rephrased to be clearer and more concise (e.g., 100-104, study design section, 307, 408-413) and also a number of spelling and grammatical mistakes throughout (e.g., 321, 401)

21 - first sentence could be rephrased ‘which got’ is weird to read? Maybe ‘and was’ or ‘which was’

54 – in what ways?

64 – ‘were also subject to’

77 – such as?

78-79 – repeat of earlier sentence

100-104 – long sentence

107 – this could be phrased more scientifically?

114 – ‘A cross-sectional, retrospective observational review of comments on specific social media recruitment posts (n=2) was undertaken’.

Study design – Second sentence is lengthy and could be rephrased

126 – advertised a research opportunity?

134 – specify these stages?

151 – correlations were used to explore? Correlations were conducted?

166 – Most data points relating to this theme highlighted a lack of support – how many? State (n= X)

174 – same as above

233 – ‘many levels’ - What other levels?

Table 3 – is this missing ‘positive emotion words’ in the top row?

307 – lengthy sentence

311 – offers insights into why recruitment to care home studies during the pandemic, using social media, may have been challenging, based on the sentiments contributed by HCWs and by the public

321 – spelling mistake ‘were > where’

322 – meaning?

343 – which in this case could mean?

377 – participants?

382 – spelling – limitations

401 – capitalised W

403 – may have been used?

405 – unsure what this means: ‘should not be considered a one-time input’

408-413 – lengthy sentence could be rephrased and split up

The influence of social media posts – not sure it shows this as much as public sentiment towards the posts

Reviewer #4: * No need to use abbreviations of authors' names in the manuscript (page 6 and elsewhere). The reader does not need to know who did what.

* The writing style often lack a terse, technical, academic writing style associated with such manuscripts. Eg. 'This got', 'To use' on page 2 and elsewhere. Later, on p. 7. "after this", 'via discussion'...

* By target population (p. 3), are the authors referring to the study's respondents?

*. Lines 97 and 98 on p. 5 need a citation to strengthen the credibility of the claim.

* P. 5, L90 - why don't HCWs have access to work email? The answer may explain the relatively low response rate and quality of the reported findings.

* Lines 121+122 (p. 6) are vague.

* Lines 332+333 make a bold claim. Were the negative consequences formally diagnosed? How? By what measures? Is there a citation to substantiate this?*

* No formal explanation provided for the low uptake claimed on line 334.

* Heading on page 18 should read Limitations of The/This Study'.

7. PLOS authors have the option to publish the peer review history of their article (what does this mean?). If published, this will include your full peer review and any attached files.

Reviewer #3: No

Reviewer #4: No

---

## [Author Response · Author response to Decision Letter 1]

24 Jul 2023

Reviewer #3: comments 

Authors responses 

Abstract :The title and bits of the abstract make it sound as if the care home itself (including residents) are the focus of the research (and the research that this sub-study is situated in)

Thank you for this comment. We have re-worded the title and the abstract to be more precise. New Title: 

Exploring barriers to care home research recruitment during the Covid-19 pandemic. The influence of social media recruitment posts and public sentiment. 

The paper itself is more about public sentiment to care home staff.

Thank you for your comment. We have explored the influence of public sentiment on research recruitment which is clearly stated in the title and the abstract. 

I became confused about the post 1 and post 2, and initially thought it was related to the two aims.

Thank you. In the last paragraph of the introduction, we have added the signposting of the two recruitment posts to clarify this. Also, in methods of the abstract, we highlighted that analysis is from 2 Facebook posts.

Not clear who the participants are in this study.

Thank you. This has now been clarified in the study design section p7.line 125. ‘Made by the public’ 

Background –

It becomes clear that this study is situated in a wider study on the mental health and wellbeing of care home staff. This revelation was a bit late as I initially thought it was about recruiting care homes to research (for residents)

Thank you. We have now clarified this in the abstract under methods.

This cross-sectional study analysed comments from two Facebook posts (available June-October 2021) advertising a study on psychological support for care staff during the pandemic. The study was situated within a larger investigation into the mental health and wellbeing of care home staff and employed both qualitative analysis and quantitative methods (word count and correlations between words used and between posts). 

Page 5, line 87, you talk about the fact that its difficult to recruit care homes to research. This is true, but there are some good initiatives out there such as the ENRICH network that are worth mentioning. You do talk about this again in your conclusion.

Thank you for this. We agree there are some good initiatives such as ENRIH network, and we have added this information in this section.

‘…in addition to some UK initiatives such as ENRICH network .’

Methods – I’m not convinced that you have really done a thematic analysis as considered by Braun and Clark. There are about 200 data items in total. This sounds more like a content analysis than a thematic analysis. Have a look at later writing eg Braun and Clarke 2021. As B&C would say, ‘themes don’t emerge’

Thank you. We have made some changes to reflect this. We replaced emerged with identified. However, we did not have predefined categories which is a prerequisite for content analysis. We coded, categorised and interpreted the data over the six step approach, which lead us to identifying the themes which fits with thematic analysis. 

I think the findings/themes developed are interesting but in order to address what it says in the title, but you need to discuss how these themes act as barriers to recruitment more.

Thank you. We agree with this statement. Explicit connections have now been made between the themes and the research questions. This results in merging of the vaccine and mistrust theme as the issues with vaccine were also linked to mistrust. Changes also made in discussion over the pages 10-13 and 18-20.

In the methods it’s still not clear who the participants are. By the results this has become clearer that it is likely primarily to be members of the UK public (particularly to post 1). I don’t think this is tidy enough.

Thank you. We corrected this.Please see response to point 4 & 5 above for changes.

Don’t you need ethical approval for this kind of research? Or at least some discussion of ethical issues? See this for example. https://www.gla.ac.uk/media/Media_487729_smxx.pdf

Thank you. Ethical approval was granted from the Faculty of Health and Life Sciences at Northumbria University REF:32662NU and consent was waived.

I get the sense that this paper was based on getting some interesting reflections back from your recruitment method and thus a bit of an additional publication.

Thank you for your comment. We had an objective to report on social media recruitment due to challenges with recruiting for research during the pandemic. 

Results – I thought there was some interesting data in here. The reflections on mistrust of researchers and timing with the required vaccination of English (not Welsh, not sure about Scotland) care home staff to have a COVID-19 vaccination are interesting.

Thank you

The quantitative analysis seems less useful. I’m not sure what the point of comparing +ve and -ve words for post 1 and 2 are, if you don’t know who your respondents are.

Thank you. While it is true we do not know exactly who the respondents are, Linguistic inquiry word count and comparison in public discourse on social media has been used in recent years to help with the understanding or predictability of mood and behaviour models. This can give us an indication/ insight in the understanding of public sentiment in similar situations (recruitment adverts). 

I’m not sure the word count analysis aspect added very much to the paper because there is no way to confidently attribute the number of times a word is said with whether it presents as a barrier to recruitment – but this is mentioned in their limitations.

Thank you for your comment. Linguistic inquiry word count in public discourse on social media has been used in recent years to help with the understanding or predictability of mood and behaviour models. This is linked with the locus of control theory, which looks at the extent to which people believe they control their lives. 

What this means for barriers to recruitment is that those that feel they have less control over their life, they are more likely to have mistrust towards government and researchers and therefore less likely to participate in research. Thus, as mentioned before, doing work that can increase trust in researchers by building relationships with care homes, can change the perception HCWs have towards researchers. 

Discussion and conclusion

I do agree that we need better collaboration between care home staff and researchers.

Thank you for your comment, we appreciate this. 

We still don’t know what the results from the survey reported. Was this method of recruitment ultimately successful or not. Your last sentence of the conclusion suggest it was, but we have no evidence of that.

Thank you for your comment. This information is now in the discussion on p18 and just before support.

You reflect on the lack of knowledge of who is responding to the post, but I think this is a more fundamental problem. Those people who reply to facebook adverts with text rather than just a click to ‘find out more’ are more likely to be confrontational, and you know nothing about their background.

Thank you. We agree that that could be the case, hence we have put that in the limitations section. 

Do we know if facebook advertising is more successful at recruiting care home staff than any other section of the society?

We found that Facebook advertising for recruitment of care staff was very useful as we used multiple approaches and this approach yielded best results. 

Overall style

There are a number of lengthy sentences that could be rephrased to be clearer and more concise (e.g., 100-104, study design section, 307, 408-413) and also a number of spelling and grammatical mistakes throughout (e.g., 321, 401)

Thank you. We have reviewed and corrected the named sections.

21 - first sentence could be rephrased ‘which got’ is weird to read? Maybe ‘and was’ or ‘which was’

This was changed to ‘was’

54 – in what ways?

We are not clear exactly what the question is here, however we have restructured the last sentence there to be clearer.

64 – ‘were also subject to’

77 – such as?

We are not sure what these re referring to?

78-79 – repeat of earlier sentence

Thank you for the comment. The two sentences were referring to two different points, but we have added ‘another ‘ to clarify the point.

100-104 – long sentence

Thank you. We restructured the sentence to read better and clearer.

107 – this could be phrased more scientifically?

Thank you. We added reference here. 

114 – ‘A cross-sectional, retrospective observational review of comments on specific social media recruitment posts (n=2) was undertaken’.

Thank you. This has been corrected.

Study design – Second sentence is lengthy and could be rephrased

Thank you. This has been broken down and corrected.

126 – advertised a research opportunity?

Thank you for your comment. We have inserted a research opportunity in the suggested sentence on p.7 

134 – specify these stages?

Thank you for your comment. We couldn’t understand which stages you wanted us to specify. Nonetheless we tried to clarify the sentence by adding Facebook to the post 1 and post 2

151 – correlations were used to explore? Correlations were conducted?

Thank you. This has been amended to ‘correlations were conducted’

166 – Most data points relating to this theme highlighted a lack of support – how many? State (n= X)

Thank you. We have added n=20 for the most data points that related to lack of support. 

174 – same as above

Thank you. We have added n=55 

233 – ‘many levels’ - What other levels?

Thank you. We have clarified this and now reads: Blame was palpable at multiple levels (care home staff, NHS, Government and academia) and on occasions were extreme and expressed as seeking vengeance

ble 3 – is this missing ‘positive emotion words’ in the top row?

Thank you. This is not missing positive emotion in top left corner. All correlations are correctly displayed. 

307 – lengthy sentence

Thank you. This has now been amended with semicolons to break up the sentence.

Negative emotion word use also positively correlated as follows: with words associated with support (r = .27, p = .03); the COVID-19 pandemic (r = .38, p <.001) in post 2; and words associated with vaccines in post 1 (r = .18, p = .04).

311 – offers insights into why recruitment to care home studies during the pandemic, using social media, may have been challenging, based on the sentiments contributed by HCWs and by the public

Thank you. We have now amended. Taken together, our findings also offer some insights into why recruitment to care home studies during the pandemic, using social media may have been challenging based on the sentiments shared both by HCWs and by the public.

321 – spelling mistake ‘were > where’

Thank you. This has now been amended to where.

322 – meaning?

Thank you. We have added a paragraph to this now on page 19 to clarify. 

343 – which in this case could mean?

Thank you for this. We have now restructured this paragraph and added additional information. Also, vaccine and mistrust themes were merged, so the discussion reflects this.

377 – participants?

Thank you. We have replaced this with ‘respondents’ 

382 – spelling – limitations

Thank you. We corrected this now

401 – capitalised W

Thank you. We couldn’t find where we needed o change and use capital W

403 – may have been used?

Thank you for your comment. We couldn’t find on the named line what this refers to.

405 – unsure what this means: ‘should not be considered a one-time input’

Thank you. We have clarified this now. .. should not be considered as a one – off strategy for recruitment but built in as an integral avenue for recruitment to research with appropriate ethics approval especially with hard to reach groups.

408-413 – lengthy sentence could be rephrased and split up

Thank you. We have restructured this now. 

The influence of social media posts – not sure it shows this as much as public sentiment towards the posts

Thank you. We have said in the limitation the following: Finally, we want to acknowledge that the method used is insufficient to address aim 1 entirely and future qualitative studies should be undertaken to explore this further. Aims are: 1) gain understanding of potential barriers to recruitment of HCWs in UK care homes, and 2) explore public sentiment towards care home research and care homes in the context of the COVID-19 pandemic.

Reviewer #4: comments 

Authors responses 

* No need to use abbreviations of authors' names in the manuscript (page 6 and elsewhere). The reader does not need to know who did what.

Thank you for your comment. Precise reporting was requested during an earlier round of peer review. We believe the use of abbreviations to precisely convey each authors contribution to the conduct of the study enhances the rigour and trustworthiness of the study. We therefore hope you can understand our decision to leave such reporting unchanged. 

* The writing style often lack a terse, technical, academic writing style associated with such manuscripts. Eg. 'This got', 'To use' on page 2 and elsewhere. Later, on p. 7. "after this", 'via discussion'...

Thank you for highlighting instances of where academic writing style could be strengthened and we have amended accordingly throughout. 

* By target population (p. 3), are the authors referring to the study's respondents?

We have restructured the sentence . We hope this provides clarity to a naïve reader that we are referring to codesign approaches in the research population rather than study respondents. 

*. Lines 97 and 98 on p. 5 need a citation to strengthen the credibility of the claim.

Thank you. We have added the following citation to strengthen this:

Tsao, S. F., Chen, H., Tisseverasinghe, T., Yang, Y., Li, L., & Butt, Z. A. (2021). What social media told us in the time of COVID-19: a scoping review. The Lancet Digital Health, 3(3), e175-e194.

* P. 5, L90 - why don't HCWs have access to work email? The answer may explain the relatively low response rate and quality of the reported findings.

Thank you. Prior research has illustrated this as a potential barrier to study recruitment. However, the explanatory mechanism for this barrier remains underexplored; and yes, we would agree merits further exploration. We have thus restructured relevant sentences. 

* Lines 121+122 (p. 6) are vague.

Thank you. We have restructured this line and hope this provides the necessary clarity. 

* Lines 332+333 make a bold claim. Were the negative consequences formally diagnosed? How? By what measures? Is there a citation to substantiate this?*

Thank you for highlighting this sentence. We have revised this sentence (and the following sentences) accordingly to make this claim more tentative. However, we believe emergent evidence cited in the follow up sentence supports claims made. 

* No formal explanation provided for the low uptake claimed on line 334.

Thank you. We agree that understanding explanatory mechanisms unpicking the low uptake of well-being interventions such as PFA during the COVID-19 pandemic is merited. However, the nascent evidence base to our knowledge has not yet reported explanatory mechanisms for this phenomenon. The need for further research to understand these explanatory mechanisms is reported in the contemporaneous literature 1

1. Schoultz M, McGrogan C, Beattie M, Macaden L, Carolan C, Dickens G. Uptake and Effects of Psychological First Aid Training For Healthcare Workers’ Wellbeing in Nursing Homes: A UK National Survey.PLOS One. 2022 Nov 17(11) https://doi.org/10.1371/journal.pone.0277062

* Heading on page 18 should read Limitations of The/This Study'.

Thank you for your suggestion. The submission guide is not specific on this . In addition, if we change the heading to limitations only and omit strengths while explicitly discussing strengths and limitations in the section will incorrectly signpost the reader for this section.

---

## [Decision Letter · Decision Letter 2]

10 Nov 2023

PONE-D-22-20731R2Exploring barriers to care home research recruitment during the Covid-19 pandemic. The influence of social media recruitment posts and public sentiment.PLOS ONE

Dear Dr. Schoultz,

Thank you for submitting your manuscript to PLOS ONE. After careful consideration, we feel that it has merit but does not fully meet PLOS ONE’s publication criteria as it currently stands. Therefore, we invite you to submit a revised version of the manuscript that addresses the points raised during the review process.

We look forward to receiving your revised manuscript.

Kind regards,

Ali B. Mahmoud, Ph.D.

Academic Editor

PLOS ONE

Journal Requirements:

Reviewers' comments:

Reviewer's Responses to Questions

**Comments to the Author**

1. If the authors have adequately addressed your comments raised in a previous round of review and you feel that this manuscript is now acceptable for publication, you may indicate that here to bypass the “Comments to the Author” section, enter your conflict of interest statement in the “Confidential to Editor” section, and submit your "Accept" recommendation.

Reviewer #4: (No Response)

2. Is the manuscript technically sound, and do the data support the conclusions?

Reviewer #4: Partly

3. Has the statistical analysis been performed appropriately and rigorously? 

Reviewer #4: Yes

4. Have the authors made all data underlying the findings in their manuscript fully available?

Reviewer #4: Yes

5. Is the manuscript presented in an intelligible fashion and written in standard English?

Reviewer #4: No

6. Review Comments to the Author

Reviewer #4: This manuscript has several shortcomings that require consideration. These shortcomings include, but are not limited to:

1. There is no theoretical lens driving the study. The addition of a relevant theoretical lens adds to the credibility of the study, and improves the methodology and findings.

2. Grammar is problematic throughout the manuscript. E.g. 'care home' (in the title and elsewhere). Do the authors mean 'home care'? 'Research recruitment' (also in the title). Do the authors mean 'recruitment research'? '... recruitment to home care research...' (p. 3). Etc.

3. In the abstract, the introduction starts with an unsubstantiated bold claim. Where is the evidence to support this?

4. In the same paragraph the authors claim that social media is 'becoming'... No. It's already there.

5. The entire aim (on p. 2) of the study rests on the analysis of two social media posts. The authors make do not address the glaring possibility that this may not be a large enough sample size.

6. How are the findings 'novel'? (p. 3)

7. The authors reference a government twice in the first paragraph in the introduction. It's not until the next paragraph we find they are referring to the English government.

8. The authors assert the negative media portrayal of 'care home staff' without an explanation to why this negative media portrayal exists. This is important in understanding the three themes stated in the research.

9. Why are 19-21 year olds 'hard to reach'? How do we know this?

10. The authors refer to MS on p. 7. Presumably this is the name of one of the authors. Given that the use of technology is being discussed in this section, MS could be an abbreviation for Microsoft.

11. In one of the more glaring omissions in the reviewed manuscript is the use of constructs, referred to in the manuscript as themes, namely support, mistrust, and blame. Yet none of these are defined in the context of the study. E.g. Support in the health care industry can take several forms, for instance physical support, emotional support, etc. These constructs merit precise definition which assists in better understanding the data analysis.

7. PLOS authors have the option to publish the peer review history of their article (what does this mean?). If published, this will include your full peer review and any attached files.

Reviewer #4: No

---

## [Author Response · Author response to Decision Letter 2]

19 Dec 2023

Dear Reviewer

We thank you for all your suggestions. please see our responses below

Reviewers comments 

1. There is no theoretical lens driving the study. The addition of a relevant theoretical lens adds to the credibility of the study, and improves the methodology and findings. 

Thank you for your valuable comment. We have incorporated Health Belief Model (HBM) in the manuscript to help contextualize our research within the broader literature and guide the interpretation of our findings. Please see changes on page 4.

2. Grammar is problematic throughout the manuscript. E.g. 'care home' (in the title and elsewhere). Do the authors mean 'home care'? 'Research recruitment' (also in the title). Do the authors mean 'recruitment research'? '... recruitment to home care research...' (p. 3). Etc. 

Thank you for your comment. 

Regarding the term 'care home,' we acknowledge that terminology can vary, and we understand the potential confusion. In the context of our study, 'care home' refers to residential facilities where individuals, particularly the elderly, receive assisted living and health and social care services. This is widely used term in UK.

Similarly, we note your observation regarding the phrase 'research recruitment.' We intended to convey the process of recruiting participants for a research study. Again, this is widely used terminology in UK.

3. In the abstract, the introduction starts with an unsubstantiated bold claim. Where is the evidence to support this? 

Thank you for your comment. The evidence for this is referenced in the main text page 5:

Law E, Ashworth R. Facilitators and Barriers to Research Participation in Care Homes: Thematic Analysis of Interviews with Researchers, Staff, Residents and Residents’ Families. J Long-Term Care [Internet]. 2022 Feb 23 [cited 2022 Jul 21];2022:49–60. 

Moore DC, Payne S, Van Den Block L, Ten Koppel M, Szczerbińska K, Froggatt K. Research, recruitment and observational data collection in care homes: Lessons from the PACE study. BMC Res Notes. 2019 Aug 14;12(1).

4. In the same paragraph the authors claim that social media is 'becoming'... No. It's already there. 

Thank you. We have changed this to: has increased since Covid-19

5. The entire aim (on p. 2) of the study rests on the analysis of two social media posts. The authors make do not address the glaring possibility that this may not be a large enough sample size. 

Thank you for your comment. We appreciate your concern regarding the sample size and its potential impact on the robustness of our study findings. We recognize the importance of addressing this consideration explicitly.

The focus of our study on the analysis of two social media posts is indeed a limitation, and we have acknowledged the potential impact on the generalizability of our findings. While our intent was to gain valuable insights into public sentiment and potential barriers to recruitment based on these posts, we have made sure the limitation has been discussed explicitly on page 23.

6. How are the findings 'novel'? (p. 3) 

Thank you. Our study makes several contributions to the existing literature. Firstly, we employed a mixed-methods approach, combining thematic analysis with quantitative exploration of word usage. This hybrid approach provides a comprehensive understanding of public sentiment on social media, which goes beyond a purely qualitative or quantitative analysis.

The thematic analysis revealed three prominent themes: support, mistrust, and blame. These themes reflect the complex and multifaceted nature of public discourse during the COVID-19 pandemic, especially in the context of care home workers. Our findings shed light on the nuanced perspectives and emotions expressed by social media users, capturing the depth of their experiences and concerns.

The quantitative analysis, specifically the word count and t-tests, allowed us to quantify and compare the prevalence of certain words and themes between the two Facebook posts. This approach provides a structured and objective dimension to our findings, enhancing the depth of our analysis.

Moreover, the correlations presented in the study offer insights into the relationships between different word categories, revealing potential patterns and connections in the public discourse. This nuanced exploration of language use contributes to a richer understanding of the sentiments expressed in the social media comments.

In summary, the novelty of our findings lies in the integration of qualitative and quantitative methods, the identification of themes that reflect the complexity of public sentiment, and the exploration of word usage patterns that add depth to our understanding. We believe that our study provides a valuable contribution to the field, offering insights into the public discourse surrounding care home workers during the pandemic.

7. The authors reference a government twice in the first paragraph in the introduction. It's not until the next paragraph we find they are referring to the English government. 

Thank you for your comment. The introduction specifies UK national governments page 4 9see sentence below). The study was open to whole of UK and therefore its not specifically English government. 

‘Additionally, the perceived lack of support from the UK national governments further heightened the sense of vulnerability among HCWs, with a substantial proportion considering leaving their professions due to the inadequacies of support systems (2). ‘

8. The authors assert the negative media portrayal of 'care home staff' without an explanation to why this negative media portrayal exists. This is important in understanding the three themes stated in the research. 

Thank you. We acknowledge the importance of addressing the origins of negative media portrayals to enhance the comprehensiveness of our study. In the revised manuscript, we have included a paragraph that explores potential reasons for the negative portrayal of care home staff in the media. The negative media portrayal of care homes during the COVID-19 pandemic can be attributed to several interconnected factors. Heightened mortality rates within care homes, often resulting from the virulent nature of the virus, have become focal points in media coverage, creating a perception of inadequacy in crisis management. Reports on insufficient protective measures, including shortages of essential equipment, and staffing challenges, such as shortages and overwork, contribute to the negative image. Communication issues, both within care homes and between stakeholders, are highlighted, further impacting the perception of care home efficacy. Public and governmental responses, including policies and financial support, are scrutinized, adding another layer to the narrative. Stigmatization of care homes as COVID-19 hotspots and potential sensationalism by the media contribute to a nuanced and often negative portrayal. This complex interplay of factors underscores the multifaceted nature of the media's influence on public perception during the pandemic. Page 23/24.

---

## [Decision Letter · Decision Letter 3]

5 Feb 2024

PONE-D-22-20731R3Exploring barriers to care home research recruitment during the Covid-19 pandemic. The influence of social media recruitment posts and public sentiment.PLOS ONE

Dear Dr. Schoultz,

Thank you for submitting your manuscript to PLOS ONE. After careful consideration, we feel that it has merit but does not fully meet PLOS ONE’s publication criteria as it currently stands. Therefore, we invite you to submit a revised version of the manuscript that addresses the points raised during the review process.

We look forward to receiving your revised manuscript.

Kind regards,

Ali B. Mahmoud, Ph.D.

Academic Editor

PLOS ONE

Reviewers' comments:

Reviewer's Responses to Questions

**Comments to the Author**

1. If the authors have adequately addressed your comments raised in a previous round of review and you feel that this manuscript is now acceptable for publication, you may indicate that here to bypass the “Comments to the Author” section, enter your conflict of interest statement in the “Confidential to Editor” section, and submit your "Accept" recommendation.

Reviewer #5: (No Response)

2. Is the manuscript technically sound, and do the data support the conclusions?

Reviewer #5: Partly

3. Has the statistical analysis been performed appropriately and rigorously? 

Reviewer #5: Yes

4. Have the authors made all data underlying the findings in their manuscript fully available?

Reviewer #5: Yes

5. Is the manuscript presented in an intelligible fashion and written in standard English?

Reviewer #5: No

6. Review Comments to the Author

Reviewer #5: This paper provides a mixed methods assessment of responses to 2 posts made on Facebook to recruit care home staff for a UK-wide research survey study during the COVID-19 pandemic. It includes qualitative thematic analysis and word categorization and count analysis.

The responses to the recruitment posts are an interesting sample for determining public and care home staff perspectives to the study and the pandemic more broadly.

The 2 main research questions are to “gain understanding of potential barriers to study recruitment of HCWs in UK care homes, and to explore the public sentiment towards care home research and care homes in the context of the COVID-19 pandemic.”

The thematic analysis methods are well described and appropriate for the type of data. Three themese were identified: support, mistrust, and blame. The hypotheses for comparing the length and word category usage between the posts is not clear. Having supporting references for this approach and the meaning that can be derived from the analysis would be useful and likely strengthen the interpretation.

Major Comments

-----------------

1. the authors have added in the Health Belief Model in the intro without first describing it and how it is relevant. How does it provide a valuable lens?

2. Were the word categories and bank derived from the study team? Did you seek lexicons to use? What are the hypotheses related to first, second, third person pronouns?

3. For a study described as mixed methods, the authors do not describe commonalities or divergence between the 2 methods (except for support which overlaps as a word category and a theme). For example, in the discussion “With attribution to blame, there was a larger range of 356 negative emotion words than positive emotion words.” Where is the analysis to support this?

4. “Findings suggest that care home staff received little psychological support during the pandemic…” Where is the supporting evidence for this in the results? Does the theme of support only refer to psychological support? It sounds like other kind of support are covered under the theme.

5. Could you always tell if a response was by a HCW?

6. How does HBM support the findings? There is no mention of it in the discussion.

Minor Comments

-----------------

1. There are several typos throughout, starting in the abstract (extra spaces in line 27, commas in line 28). Others: line 66, 73, 120; grammar in lines 207-208; 386 -missing period; 389 -should be experiences; 472; 479 that should replace than

2. In the abstract, details about the 2 posts are needed because the results provided have no context. Was it just timing of the posts? Content of the posts?

3. Define what you mean by ‘choice words’

4. Is ‘with’ missing after co-design in line 55?

5. What is the limitation of not knowing reach for post 1 (lines 181-2)? Especially related to number of responses to the posts.

6. References needed for lines 482-502

7. PLOS authors have the option to publish the peer review history of their article (what does this mean?). If published, this will include your full peer review and any attached files.

Reviewer #5: No

---

## [Author Response · Author response to Decision Letter 3]

18 Apr 2024

Dear reviewer,

Thank you for your valuable feedback and insightful questions. we have addressed each feedback below and we believe that has significantly improved the manuscript: 

1. the authors have added in the Health Belief Model in the intro without first describing it and how it is relevant. How does it provide a valuable lens?

Thank you for this comment. We have added the following to clarify the lens of HBM 

In this context, HCWs' perception of the threat posed by the virus, both to their physical well-being and mental health, aligns with the core tenets of the HBM such as perceived susceptibility, perceived severity, perceived benefits and barriers, cues to action and self- efficacy which combined shape the individuals health beliefs and influence their health-related decision making. 

2. Were the word categories and bank derived from the study team? Did you seek lexicons to use? What are the hypotheses related to first, second, third person pronouns? 

Thank you for your valuable feedback and insightful questions regarding our study's methodology and hypotheses. We appreciate the opportunity to address these points in more detail.

Regarding the derivation of word categories and the word bank, they were indeed derived from the findings of our thematic analysis conducted by the study team. We systematically identified recurring themes and concepts from the qualitative data, which informed the creation of word categories for the quantitative analysis. We did not specifically seek external lexicons for this purpose, as our aim was to develop categories tailored to the specific context of our study.

Regarding the hypotheses related to first, second, and third person pronouns, our analysis was guided by the following considerations:

Perceptions of Personal Responsibility: We hypothesized that the frequency of first-person pronouns (e.g., "I," "me," "my") used in the comments would reflect individuals' perceptions of personal responsibility for their actions and behaviours during the pandemic. Higher usage of first-person pronouns may indicate a stronger sense of personal accountability.

Displacement of Responsibility: We expected that the use of second person pronouns (e.g., "you," "your") might suggest attempts to shift responsibility onto others or external factors. This displacement of responsibility could manifest in statements that attribute blame or emphasise external influences on behaviour.

Locus of Control: We anticipated that the frequency of third person pronouns (e.g., "he," "she," "they") could provide insights into individuals' locus of control regarding the pandemic. Higher usage of third person pronouns may suggest a perception that events are influenced by external forces beyond personal control.

These hypotheses guided our quantitative analysis of word usage to explore perceptions of personal responsibility, displacement of responsibility, and locus of control among participants in the context of the pandemic. We have included additional clarification on these points in the revised manuscript.

3. For a study described as mixed methods, the authors do not describe commonalities or divergence between the 2 methods (except for support which overlaps as a word category and a theme). For example, in the discussion “With attribution to blame, there was a larger range of 356 negative emotion words than positive emotion words.” Where is the analysis to support this? 

Thank you for this comment. These are the steps in the analysis we took: 

Quantitative Analysis of Word Usage: Quantified the frequency of positive and negative emotion words used in each post. We manually coded the comments to identify and count words associated with positive and negative emotions. This quantitative analysis provided empirical evidence to support the statement that there was a larger range of negative emotion words than positive emotion words in comments attributed to blame.

Comparative Analysis of Themes and Word Categories: Conducted a comparative analysis between the themes identified from the qualitative data and the word categories derived from the quantitative analysis. Looked for commonalities or discrepancies between the themes and word categories to provide a more integrated understanding of the data. For example, we explored how the theme of blame correlated with the frequency of negative emotion words identified in the word categories.

Integration of Qualitative and Quantitative Findings: we Integrated the qualitative findings with the quantitative data to provide a more nuanced interpretation of the results. Discussed how the quantitative analysis of word usage complements or expands upon the qualitative themes identified in the study. For instance, we elaborated on how the quantitative analysis of negative emotion words reinforces the qualitative theme of blame and its implications for understanding public sentiments towards research in care homes during the pandemic

4. “Findings suggest that care home staff received little psychological support during the pandemic…” Where is the supporting evidence for this in the results? Does the theme of support only refer to psychological support? It sounds like other kind of support are covered under the theme. 

Thank you for your insightful feedback and question regarding the evidence supporting the assertion about psychological support for care home staff during the pandemic and the interpretation of the theme of support in our study.

In our analysis, the theme of support encompasses various forms of support, including but not limited to psychological support. While our discussion highlights the potential lack of psychological support for care home staff during the pandemic, we acknowledge that other forms of support, such as logistical support, social support, and access to resources, are also critical for their well-being.

The evidence supporting the assertion about psychological support for care home staff is derived from several sources within our study:

Qualitative Analysis of Themes: The theme of support emerged from our thematic analysis of the comments on social media recruitment posts. While the theme encompasses different types of support, the discussion in the manuscript focuses primarily on the potential lack of psychological support, as indicated by the greater use of negative emotion words in comments from care home staff (post 2) compared to those from the general public (post 1).

Word Count Analysis: Our quantitative analysis of word usage revealed a higher frequency of words associated with support and negative emotions in comments from care home staff (post 2), suggesting a potential lack of psychological support. Additionally, the positive relationship between words associated with support and negative emotive words, as well as first person singular pronouns, in post 2 comments further supports the notion that care home staff perceived a lack of support, which may have contributed to negative emotional experiences.

Literature Review: The assertion is also supported by previous research (citations 11, 25, 26-28) that has documented low uptake and awareness of well-being interventions among healthcare workers, including care home staff, during the pandemic, as well as increased rates of depression, anxiety, and insomnia among this population.

5. Could you always tell if a response was by a HCW? 

Thank you for your question regarding the ability to distinguish responses from healthcare workers (HCWs) within our study.

In our analysis, we did not have direct identifiers for individuals' occupations or roles. Therefore, we were unable to definitively determine whether a response came from an HCW based solely on the content of the comments. We acknowledged this in our limitations. However, we employed several strategies to infer the likelihood of a response being from an HCW:

Contextual Clues: We considered contextual factors such as the topic of discussion and the post being directly addressing HCWs (i.e., healthcare-related issues, experiences in care homes), which may suggest the involvement of HCWs. Comments discussing firsthand experiences in healthcare settings or referencing specific challenges faced by HCWs during the pandemic were likely to be from individuals working in healthcare.

Language and Terminology: We examined the language and terminology used in the comments, including technical medical terminology, professional jargon, or references to specific healthcare procedures or protocols. While not definitive, the presence of such language may indicate that the commenter has a background in healthcare.

Self-Identification: Some commenters may explicitly identify themselves as HCWs or mention their profession or role in their comments. We took these self-identifications into account when analysing the data.

Analysis of Pronoun Use: As mentioned in our study findings, we conducted an analysis of pronoun use, including the frequency of first-person singular pronouns (e.g., "I," "me," "my"), which may reflect personal experiences and perspectives. Higher usage of such pronouns may suggest that the commenter is sharing their own experiences as an HCW.

6. How does HBM support the findings? There is no mention of it in the discussion. 

Thank you for this. We have added to the discussion to address your feedback.

While our analysis primarily focused on themes of support, mistrust, and blame, integrating insights from the Health Belief Model (HBM) enriches our understanding of the underlying mechanisms driving these sentiments. (p17)

Our findings underscore the crucial role of support, encompassing both psychological and logistical assistance, in mitigating the adverse effects of the pandemic on care home staff's well-being. The Health Belief Model provides a valuable framework for interpreting these findings, particularly regarding perceptions of susceptibility, severity, and perceived benefits of seeking support. Perceived susceptibility to negative outcomes, such as mental health challenges, may motivate individuals to seek support services. However, our study suggests that care home staff received little psychological support during the pandemic, aligning with previous research demonstrating low uptake of well-being interventions among HCWs (11,25). This perceived lack of support may exacerbate feelings of vulnerability and contribute to increased depression, anxiety, and insomnia (26-28). The positive relationship between words associated with support and negative emotive words in our word count analysis further underscores the importance of addressing support deficits to alleviate psychological distress among care home staff.(p18-19)

Applying the Health Belief Model, individuals' perceptions of blame may reflect their beliefs about the causes and controllability of adverse outcomes. (p22)

Minor comments

1. There are several typos throughout, starting in the abstract (extra spaces in line 27, commas in line 28). Others: line 66, 73, 120; grammar in lines 207-208; 386 -missing period; 389 -should be experiences; 472; 479 that should replace than 

Thank you for pointing these out. These have happened as a result to tracked changes which have now been corrected and the rest of the manuscript checked for further errors. Some of these lines have moved and are now 210; 405;408;493 and 500.

2. In the abstract, details about the 2 posts are needed because the results provided have no context. Was it just timing of the posts? Content of the posts? Thank you for this comment. The following additions provide additional context regarding the timing and rationale behind using social media for recruitment, addressing the reviewer's comment about the need for more details on the context of the two Facebook recruitment posts.

The sentence added to provide context about the surge in social media use during COVID-19 lockdowns: "Social media use has surged since the onset of COVID-19 lockdowns, offering a plausible approach to understanding the barriers to care home research recruitment and gaining insight into public perceptions of care home workers."

The inclusion of the timing of the Facebook recruitment posts: "available June-October 2021"

3. Define what you mean by ‘choice words’ Thank you. The term "choice words" typically refers to specific words or phrases chosen deliberately to convey a particular meaning, tone, or emphasis in communication. These words are carefully selected to express one's thoughts, feelings, or intentions in a concise and impactful manner. In the context of analysing comments on social media posts, "choice words" may refer to words or phrases that reflect a particular sentiment, attitude, or opinion expressed by the commenter. These words can provide insights into the commenter's perspective, emotions, or intentions regarding the topic being discussed. 

4.Is ‘with’ missing after co-design in line 55? 

Thank you. We added the word ‘with’ after co-design. 

5. What is the limitation of not knowing reach for post 1 (lines 181-2)? Especially related to number of responses to the posts. 

Thank you. The limitation of not knowing the reach for post 1, particularly in relation to the number of responses, is noteworthy. However, it's important to acknowledge that despite the absence of specific reach data for post 1, we believe its reach was likely wider than post 2. This assumption is based on the understanding that post 1 was advertised on the official university page, which typically has a larger and more diverse audience compared to individual or smaller community pages.

6. References needed for lines 482-502 

Thank you. We have added the following references: 

36. Tausczik, Yla R., and James W. Pennebaker. "The psychological meaning of words: LIWC and computerized text analysis methods." Journal of language and social psychology 29, no. 1 (2010): 24-54.

37. Kramer, Adam DI, Jamie E. Guillory, and Jeffrey T. Hancock. "Experimental evidence of massive-scale emotional contagion through social networks." Proceedings of the National academy of Sciences of the United States of America 111, no. 24 (2014): 8788.

38. Miller, Edward Alan, Elizabeth Simpson, Pamela Nadash, and Michael Gusmano. "Thrust into the spotlight: COVID-19 focuses media attention on nursing homes." The Journals of Gerontology: Series B 76, no. 4 (2021): e213-e218.

39. Giri, Shamik, Lee Minn Chenn, and Roman Romero-Ortuno. "Nursing homes during the COVID-19 pandemic: a scoping review of challenges and responses." European Geriatric Medicine 12, no. 6 (2021): 1127-1136

---

## [Decision Letter · Decision Letter 4]

29 Apr 2024

Exploring barriers to care home research recruitment during the Covid-19 pandemic. The influence of social media recruitment posts and public sentiment.

PONE-D-22-20731R4

Dear Dr. Schoultz,

We’re pleased to inform you that your manuscript has been judged scientifically suitable for publication and will be formally accepted for publication once it meets all outstanding technical requirements.

Kind regards,

Ali B. Mahmoud, Ph.D.

Academic Editor

PLOS ONE

Additional Editor Comments (optional):

Reviewers' comments:

Reviewer's Responses to Questions

**Comments to the Author**

1. If the authors have adequately addressed your comments raised in a previous round of review and you feel that this manuscript is now acceptable for publication, you may indicate that here to bypass the “Comments to the Author” section, enter your conflict of interest statement in the “Confidential to Editor” section, and submit your "Accept" recommendation.

Reviewer #5: All comments have been addressed

2. Is the manuscript technically sound, and do the data support the conclusions?

Reviewer #5: Yes

3. Has the statistical analysis been performed appropriately and rigorously? 

Reviewer #5: Yes

4. Have the authors made all data underlying the findings in their manuscript fully available?

Reviewer #5: Yes

5. Is the manuscript presented in an intelligible fashion and written in standard English?

Reviewer #5: Yes

6. Review Comments to the Author

Reviewer #5: The authors have appropriately addressed my earlier comments and the resulting version is more clear.

7. PLOS authors have the option to publish the peer review history of their article (what does this mean?). If published, this will include your full peer review and any attached files.

Reviewer #5: No

---

## [Editor Report · Acceptance letter]

26 May 2024

PONE-D-22-20731R4 

PLOS ONE

Dear Dr. Schoultz, 

I'm pleased to inform you that your manuscript has been deemed suitable for publication in PLOS ONE. Congratulations! Your manuscript is now being handed over to our production team.

Kind regards, 

on behalf of

Dr. Ali B. Mahmoud 

Academic Editor

PLOS ONE